# REVISITING THE MASTER-SLAVE ARCHITECTURE IN MULTI-AGENT DEEP REINFORCEMENT LEARNING

## ABSTRACT

Many tasks in artificial intelligence require the collaboration of multiple agents. We exam deep reinforcement learning for multi-agent domains. Recent research efforts often take the form of two seemingly conflicting perspectives, the decentralized perspective, where each agent is supposed to have its own controller; and the centralized perspective, where one assumes there is a larger model controlling all agents. In this regard, we revisit the idea of the master-slave architecture by incorporating both perspectives within one framework. Such a hierarchical structure naturally leverages advantages from one another. The idea of combining both perspectives is intuitive and can be well motivated from many real world systems, however, out of a variety of possible realizations, we highlights three key ingredients, i.e. composed action representation, learnable communication and independent reasoning. With network designs to facilitate these explicitly, our proposal consistently outperforms latest competing methods both in synthetic experiments and when applied to challenging StarCraft[1] micromanagement tasks.

## 1 INTRODUCTION

Reinforcement learning (RL) provides a formal framework concerned with how an agent takes actions in one environment so as to maximize some notion of cumulative reward. Recent years have witnessed successful application of RL technologies to many challenging problems, ranging from game playing [17; 21] to robotics [9] and other important artificial intelligence (AI) related fields such as [20] etc. Most of these works have been studying the problem of a single agent.

However, many important tasks require the collaboration of multiple agents, for example, the coordination of autonomous vehicles [2], multi-robot control [13], network packet delivery [32] and multi-player games [25] to name a few. Although multi-agent reinforcement learning (MARL) methods have historically been applied in many settings [1; 31], they were often restricted to simple environments and tabular methods.

Motivated from the success of (single agent) deep RL, where value/policy approximators were implemented via deep neural networks, recent research efforts on MARL also embrace deep networks and target at more complicated environments and complex tasks, e.g. [23; 19; 4; 12] etc. Regardless though, it remains an open challenge how deep RL can be effectively scaled to more agents in various situations. Deep RL is notoriously difficult to train. Moreover, the essential state-action space of multiple agents becomes geometrically large, which further exacerbates the difficulty of training for multi-agent deep reinforcement learning (deep MARL for short).

From the viewpoint of multi-agent systems, recent methods often take the form of one of two perspectives. That is, the decentralized perspective where each agent has its own controller; and the centralized perspective where there exists a larger model controlling all agents. As a consequence, learning can be challenging in the decentralized settings due to local viewpoints of agents, which perceive non-stationary environment due to concurrently exploring teammates. On the other hand, under a centralized perspective, one needs to directly deal with parameter search within the geometrically large state-action space originated from the combination of multiple agents.

---

[1]StarCraft and its expansion StarCraft: Brood War are trademarks of Blizzard Entertainment[TM]

In this regard, we revisit the idea of master-slave architecture to combine both perspectives in a complementary manner. The master-slave architecture is a canonical communication architecture which often effectively breaks down the original challenges of multiple agents. Such architectures have been well explored in multi-agent tasks [18; 28; 15; 16]. Although our designs vary from these works, we have inherited the spirit of leveraging agent hierarchy in a master-slave manner. That is, the master agent tends to plan in a global manner without focusing on potentially distracting details from each slave agent and meanwhile the slave agents often locally optimize their actions with respect to both their local state and the guidance coming from the master agent. Such idea can be well motivated from many real world systems. One can consider the master agent as the central control of some organized traffic systems and the slave agents as each actual vehicles. Another instantiation of this idea is to consider the coach and the players in a football/basketball team. However, although the idea is clear and intuitive, we notice that our work is among the first to explicitly design master-slave architecture for deep MARL.

Specifically, we instantiate our idea with policy-based RL methods and propose a multi-agent policy network constructed with the master-slave agent hierarchy. For both each slave agent and the master agent, the policy approximators are realized using recurrent neural networks (RNN). At each time step, we can view the hidden states/representations of the recurrent cells as the "thoughts" of the agents. Therefore each agent has its own thinking/reasoning of the situation. While each slave agent takes local states as its input, the master agent takes both the global states and the messages from all slave agents as its input. The final action output of each slave agent is composed of contributions from both the corresponding slave agent and the master agent. This is implemented via a gated composition module (GCM) to process and transform "thoughts" from both agents to the final action.

We test our proposal (named MS-MARL) using both synthetic experiments and challenging Star-Craft micromanagement tasks. Our method consistently outperforms recent competing MARL methods by a clear margin. We also provide analysis to showcase the effectiveness of the learned policies, many of which illustrate interesting phenomena related to our specific designs.

In the rest of this paper, we first discuss some related works in Section 2. In Section 3, we introduce the detailed proposals to realize our master-slave multi-agent RL solution. Next, we move on to demonstrate the effectiveness of our proposal using challenging synthetic and real multi-agent tasks in Section 4. And finally Section 5 concludes this paper with discussions on our findings. Before proceeding, we summarize our major contributions as follows

- We revisit the idea of master-slave architecture for deep MARL. The proposed instantiation effectively combines both the centralized and decentralized perspectives of MARL.
- Our observations highlight and verify that composable action representation, independent master/slave reasoning and learnable communication in-between are key factors to be successful in MS-MARL.
- Our proposal empirically outperforms recent state-of-the-art methods on both synthetic experiments and challenging StarCraft micromanagement tasks, rendering it a novel competitive MARL solution in general.

## 2    RELATED WORK

Current main stream RL methods apply conventional wisdoms such as Q-learning, policy gradient, actor-critic etc. [24]. Recent progress mainly focuses on practical adaptations especially when applying deep neural networks as value/policy approximators. [10] provides a recent review on deep RL.

Although MARL has been studied in the past, they have been focused on simple tasks [1]. Only until recently, with the encouragement from the successes of deep RL, deep MARL has become a popular research area targeting at more complex and realistic tasks, see e.g. [3; 23; 7; 19; 4; 12] etc.

[3] and [7] are among the first to propose learnable communications via back-propagation in deep Q-networks. However, due to their motivating tasks, both works focused on a decentralized perspective and usually applies to only a limited number of agents.

[27], [5] and [19] all proposed practical network structure or training strategies from a centralized perspective of MARL. Specifically, [19] proposed a bidirectional communication channel among

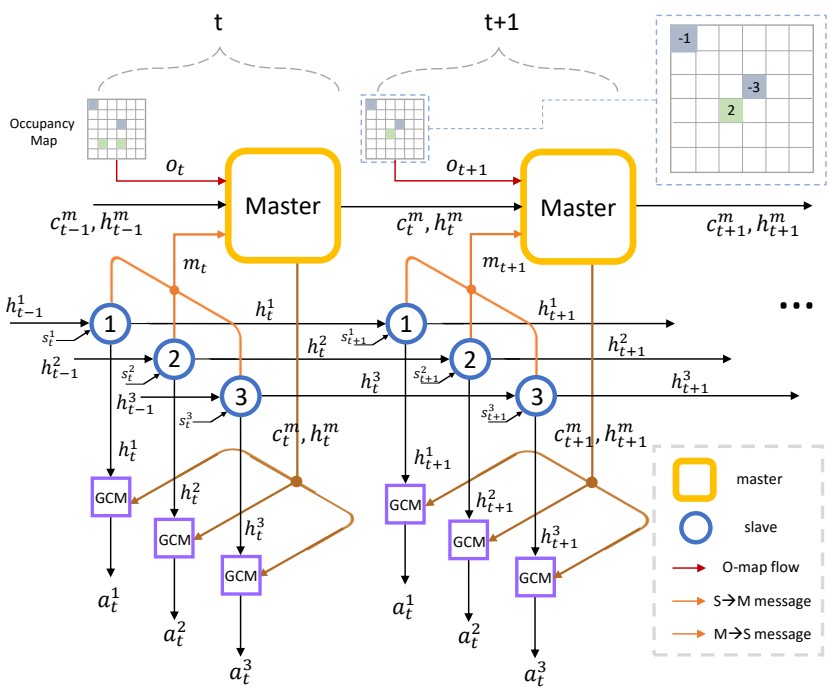

Figure 1: Pipeline of our master-slave multi-agent architecture

all agents to facilitate effective communication and many interesting designs toward the StarCraft micromanagement tasks. [27] proposed episodic exploration strategy for deterministic policy search and [5] proposed the concept of stabilizing experience replay for MARL.

Note that the above works take only one of the two perspectives and are then inherently missing out the advantages of the other. Perhaps the most related works are from [3], [4] and [12]. [23] proposed the "CommNet", where a broadcasting communication channel among all agents was set up to share global information realized as summation of the output from all individual agents. This design represents an initial version of the proposed master-slave framework, however it does not facilitate an independently reasoning master agent which takes in messages from all agents step by step and processes such information in an recurrent manner. In [4] and [12], a global critic was proposed, which could potentially work at a centralized level, however since critics are basically value networks, they do not provide explicit policy guidance. Therefore they tend to work more like a commentator of a game who job is to analyze and criticize the play, rather than a coach coaching the game.

As discussed above, the master-slave architecture has already been studied in several multi-agent scenarios. [18] utilized the master-slave architecture to resolve conflicts between multiple soccer agents; while [28; 15; 16] explored master-slave hierarchy in RL applied to load-balancing and distributed computing environments. Our proposal can be viewed as a revisit to similar ideas for deep MARL. With the proposed designs, we facilitate independent master reasoning at a global level and each slave agent thinking at a local but focused scale, and collectively achieve optimal rewards via effective communication learned with back propagation. Compared with existing works, we emphasize such independent reasoning, the importance of which are well justified empirically in the experiments. We consistently outperform existing MARL methods and achieve state-of-the-art performance on challenging synthetic and real multi-agent tasks.

Since the master-slave architecture constructs agent hierarchy by definition, another interesting related field is hierarchical RL, e.g. [8; 29]. However, such hierarchical deep RL methods studies

the hierarchy regarding tasks or goals and are usually targeting at sequential sub-tasks where the meta-controller constantly generates goals for controllers to achieve. Master-slave architecture, on the other hand, builds up hierarchy of multiple agents and mainly focuses on parallel agent-specific tasks instead, which is fundamentally different from the problems that hierarchical RL methods are concerned with.

## 3 Master-Slave Multi-Agent RL

We start by reiterating that the key idea is to facilitate both an explicit master controller that takes the centralized perspective and organize agents in a global or high level manner and all actual slave controllers work as the decentralized agents and optimize their specific actions relatively locally while depending on information from the master controller. Such an idea can be realized using either value-based methods, policy-based methods or actor-critic methods.

### 3.1 Network Architecture

Hereafter we focus on introducing an instantiation with policy gradient methods as an example, which also represents the actual solution in all our experiments. In particular, our target is to learn a mapping from states to actions $\pi_\theta(\boldsymbol{a}_t|\boldsymbol{s}_t)$ at any given time $t$, where $\boldsymbol{s} = \{s^m, s^1, ..., s^C\}$ and $\boldsymbol{a} = \{a^1, ..., a^C\}$ are collective states and actions of $C$ agents respectively and $\theta = \{\theta^m, \theta^1, ..., \theta^C\}$ represents the parameters of the policy function approximator of all agents, including the master agent $\theta^m$. Note that we have explicitly formulated $s^m$ to represent the independent state to the master agent but have left out a corresponding $a^m$ since the master's action will be merged with and represented by the final actions of all slave agents. This design has two benefits: 1) one can now input independent and potentially more global states to the master agent; and meanwhile 2) the whole policy network can be trained end-to-end with signals directly coming from actual actions.

In Figure 1, we illustrate the whole pipeline of our master-slave multi-agent architecture. Specifically, we demonstrate the network structure unfolded at two consecutive time steps. In the left part, for example, at time step $t$, the state $\boldsymbol{s}$ consists of each $s^i$ of the $i$-th slave agent and $s^m = \boldsymbol{o}_t$ of the master agent. Each slave agent is represented as a blue circle and the master agent is represented as a yellow rectangle. All agents are policy networks realized with RNN modules such as LSTM [6] cells or a stack of RNN/LSTM cells. Therefore, besides the states, all agents also take the hidden state of RNN $\boldsymbol{h}_{t-1}$ as their inputs, representing their reasoning along time. Meanwhile the master agent also take as input some information from each slave agent $\boldsymbol{c}_i$ and broadcasts back its action output to all agents to help forming their final actions. These communications are represented via colored connections in the figure.

To merge the actions from the master agent and those from the slave agents, we propose a gated composition module (GCM), whose behavior resembles LSTM. Figure 4 illustrates more details. Specifically, this module takes the "thoughts" or hidden states of the master agent $h_t^m$ and the slave agents $h_t^i$ as input and outputs action proposals $a_t^{m \to i}$, which later will be added to independent action proposals from the corresponding slave agents $a_t^i$. Since such a module depends on both the "thoughts" from the master agent and that from certain slave agent, it facilitates the master to provide different action proposals to individual slave agents. Moreover, we demonstrate in 4 that the GCM module naturally facilitates our model to generalize to cases involving heterogeneous agents. In certain cases, one may also want the master to provide unified action proposals to all agents. This could easily be implemented as a special case where the gate related to the slave's "thoughts" shuts down, which is denoted as regular MS-MARL.

### 3.2 Learning Strategy

As mentioned above, due to our design, learning can be performed in an end-to-end manner by directly applying policy gradient in the centralized perspective. Specifically, one would update all parameters following the policy gradient theorem [24] as

$$\theta \leftarrow \theta + \lambda \sum_{t=1}^{T-1} \nabla_\theta \log \pi_\theta(\boldsymbol{s}_t, \boldsymbol{a}_t) v_t \tag{1}$$

where data samples are collected stochastically from each episode $\{s_1, a_1, r_2, ..., s_{T-1},$ $a_{T-1}, r_T\} \sim \pi_\theta$ and $v_t = \sum_{j=1}^t r_t$. Note that, for discrete action space, we applied softmax policy on the top layer and for continous action space, we adopted Gaussian policy instead. We summarize the whole algorithm in Algorithm 1 as follows.

---

**Algorithm 1:** Master-Slave Multi-Agent Policy Gradient (One Batch)

---

Randomly initialize $\theta = (\theta^m, \theta^1, \ldots, \theta^C)$;
Set $\sigma = 0.05$, $R_i = 0$;
Set $s_1 =$initial state, $o_1 =$initial occupancy map, $t = 0$;
**for** *b=1,BatchSize* **do**
    **while** $s_t \neq$*terminal* **and** $t < T$ **do**
        $t = t + 1$;
        **for** *each slave agent* $i = 1 \ldots C$ **do**
            Observe local state of agent $i$: $s_t^i$;
            Feed-forward: $(h_t^i, h_t^m) = MSNet(o_t, s_t^i, h_{t-1}^i, h_{t-1}^m; \theta^i, \theta^m)$;
            Sample action $a_t^i$ according to softmax policy or Gaussian policy;
            Execute action $a_t^i$;
            Observe reward $r_t^i$ (e.g. according to (2));
            Accumulate rewards $R_i = R_i + r_t^i$;
        Observe the next state $s_{t+1}$ and occupancy map $o_{t+1}$;
    Compute terminal reward $r_{terminal}$ (e.g. using (3));
    $r_{terminal} = \begin{cases} 1 & if\ battle\ won \\ -0.2 & else \end{cases}$ ;
    $R_i = R_i + r_{terminal}, \forall i = 1 \ldots C$;
Update network parameter $\theta^m, \theta^1, \ldots, \theta^C$ using (1);

---

### 3.3 DISCUSSIONS

As stated above, our MS-MARL proposal can leverage advantages from both the centralized perspective and the decentralized perspective. Comparing with the latter, we would like to argue that, not only does our design facilitate regular communication channels between slave agents as in previous works, we also explicitly formulate an independent master agent reasoning based on all slave agents' messages and its own state. Later we empirically verify that, even when the overall information revealed does not increase per se, an independent master agent tend to absorb the same information within a big picture and effectively helps to make decisions in a global manner. Therefore compared with pure in-between-agent communications, MS-MARL is more efficient in reasoning and planning once trained.

On the other hand, when compared with methods taking a regular centralized perspective, we realize that our master-slave architecture explicitly explores the large action space in a hierarchical way. This is so in the sense that if the action space is very large, the master agent can potentially start searching at a coarse scale and leave the slave agents focus their efforts in a more fine-grained domain. This not only makes training more efficient but also more stable in a similar spirit as the dueling Q-network design [30], where the master agent works as base estimation while leaving the slave agents focus on estimating the advantages. And of course, in the perspective of applying hierarchy, we can extend master-slave to master-master-slave architectures etc.

## 4 EXPERIMENTS

### 4.1 EVALUATION ENVIRONMENTS

To justify the effectiveness of the proposed master-slave architecture, we conducted experiments on five representative multi-agent tasks and environments in which multiple agents interact with

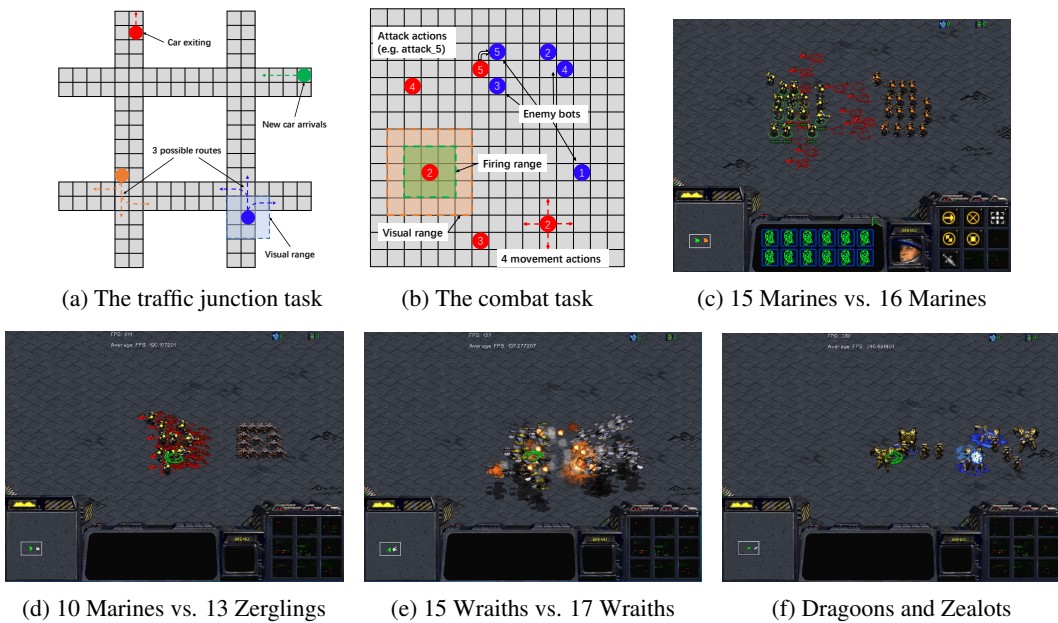

(a) The traffic junction task     (b) The combat task     (c) 15 Marines vs. 16 Marines

(d) 10 Marines vs. 13 Zerglings     (e) 15 Wraiths vs. 17 Wraiths     (f) Dragoons and Zealots

Figure 2: Environments for evaluation: (a) and (b) are established upon MazeBase Environment[22] and originally proposed in [23], (c)-(f) are mini-games from StarCraft that is included in the TorchCraft platform[25]

each other to achieve certain goals. These tasks and environments have been widely used for the evaluation of popular MARL methods such as [23; 12; 19; 27; 4; 5].

The first two tasks are the traffic junction task and the combat task both originally proposed in [23]. (Examples are shown in Figure 2(a)(b)) These two tasks are based on the MazeBase environment [22] and are discrete in state and action spaces. Detailed descriptions are given below:

**The traffic junction task** As originally designed in [23], there are three versions of this task corresponding to different hard levels. In our experiments, we choose the hardest version which consists of four connected junctions of two-way roads (as shown in Fig. 2 (a)). During every time step, new cars enter the grid area with a certain probability from each of the eight directions. Each car occupies a single-cell in the grid. All the cars keep to the right-hand side of the road. Note that there are only three possible routes for each car. A car is allowed to perform two kinds of actions at each time step: advances by one cell while keeping on its route or stay still at the current position. Once a car moves outside of the junction area, it will be removed. A collision happens when two cars move to the same location. In our experiments, following the literature, we apply the same state and reward originally designed in [23].

**The combat task** This environment simulates a simple battle between two opposing teams. The entire area is a $15 \times 15$ grid as shown in Fig. 2 (b). Both teams have 5 members and will be born at a random position. Possible actions for each agent are as follow: move one step towards four directions; attack an enemy agent within its firing range; or keep idle. One attack will cause an agent lose 1 point of health. And the initial health point of each agent is set to 3. If the health point reduces to 0, the agent will die. One team will lose the battle if all its members are dead, or win otherwise. Default settings from [23] are used in our experiments following the literature.

The following StarCraft micromanagement tasks originate from the well-known real-time strategy (RTS) game: StarCraft and was originally defined in [25; 27]. Instead of playing a full StarCraft game, the micromanagement tasks involve a local battle between two groups of units. Similar to the combat task, the possible actions also include move and attack. One group wins the game when all the units of the other group are eliminated. However, one big difference from the combat task is that the StarCraft environment is continuous in state and action space, which means a much larger search

space for learning. In other words, the combat task can also be considered as a simplified version of StarCraft micromanagement task with discrete states and actions.

The selected micromanagement tasks for our experiments are {15 Marines vs. 16 Marines, 10 Marines vs. 13 Zerglings, 15 Wraiths vs. 17 Wraiths} (shown in Figure 2 (c)-(e)). All the three tasks are categorized as "hard combats" in [19]. Thus it is quite hard for an AI bot to win the combat without learning a smart policy. On all the three tasks, independent policy methods have been proved to be less effective. Instead, MARL methods must learn effective collaboration among its agents to win these tasks e.g. [19; 27].

**15 Marines vs. 16 Marines** In this task, one needs to control 15 Terran marines to defeat a built-in AI that controls 16 Terran marines (showcased in Figure 2(a)). Note that the Marines are ranged ground units. Among the "mXvY" tasks defined in [25], such as m5v5 (5 Marines vs. 5 Marines), m10v10 (10 Marines vs. 10 Marines) m18v18 (18 Marines vs. 18 Marines), the chosen combat "15 Marines vs. 16 Marines" represents a more challenging version in the sense that the total number of agents is high and that the controlled team has less units than the enemies. Therefore a model need to learn very good strategies to win the battle. As described in [27], the key to winning this combat is to focus fire while avoiding "overkill". Besides this, another crucial policy - "Spread Out" is also captured in our experiments.

**10 Marines vs. 13 Zerglings** While we still control Marines in this scenario, the enemies belong to another type of ground unit - Zerglings. Unlike Terran Marines, Zerglings can only attack by direct contact, despite their much higher moving speed. Useful strategies also include "focus fire without overkill", which is similar to the "15 Marines vs. 16 Marines" task. Interestingly, unlike the "15M vs. 16M" task, the "Spread Out" strategy is not effective anymore in this case according to our experiments. (An example of this task is given in Figure 2 (d))

**15 Wraiths vs. 17 Wraiths** As a contrast to the above two tasks, this one is about Flying units. 15 Wraiths (ranged flying unit) are controlled to fight against 17 opponents of the same type. An important difference from the ground units is that the flying units will not collide with each other. Hence it is possible for flying units to occupy the same tile. In this case, the "Spread Out" strategy may not be important anymore. But it is essential to avoid "overkill" in this task since Wraiths have much longer "cooldown" time and much higher attack damage. (This task is shown in Figure 2 (e))

**Dragoons and Zealots** Unlike the previous three StarCraft tasks, this one involves a heterogeneous multi-agent task where 2 Dragoons(attack from remote) and 3 Zealots(attack by direct contact) need to fight against a group of enemies of the same unit type and number. Typical successful policy for this task is to focus fire on enemy zealots (low attack and high defense) at first and then eliminate the Dragoons (high attack and low defense), which was analyzed in detail by [27]. (Illustrated in Figure 2 (f))

As presented in [19], the state-of-the-art performance on these tasks are still quite low compared with others. Amazingly, the proposed MS-MARL method achieves much higher winning rates, as demonstrated in later section. We will refer to these tasks as {15M vs. 16M, 10M vs. 13Z, 15W vs. 17W and 2D3Z} for brevity in the rest of this paper.

## 4.2 IMPLEMENTATION DETAILS

In order to help reproduce our results, we hereby provide our very implementation details.

**Model Architecture** A detailed illustration of our implementation is displayed in Figure 4. As introduced in section 3, in our MS-MARL model, both the master agents and the slave agents are implemented using RNN or LSTM to incorporate temporal recurrence. Here in Figure 4, we use LSTM for the master module, and RNN for the slave module. The dimension of the hidden states in RNN or LSTM (including cell states) are all set to 50 for both the master and slave agents. The GCM module in Figure 4 (c) is noted as the "Gated Composition Module", which is introduced in section 3.1 in detail. Note that the action output is different for discrete and continuous tasks. For the traffic junction task and the combat task, the output of the network is designed as the probability of a number of actions since the action space is discrete (actually a Softmax policy). As a contrast, for "15M vs. 16M" "10M vs. 13Z" and "15W vs. 17W", our network directly generates a continuous action following Gaussian policy as described in section 3. The Softmax/Gaussian action modules are illustrated by Figure 4 D).

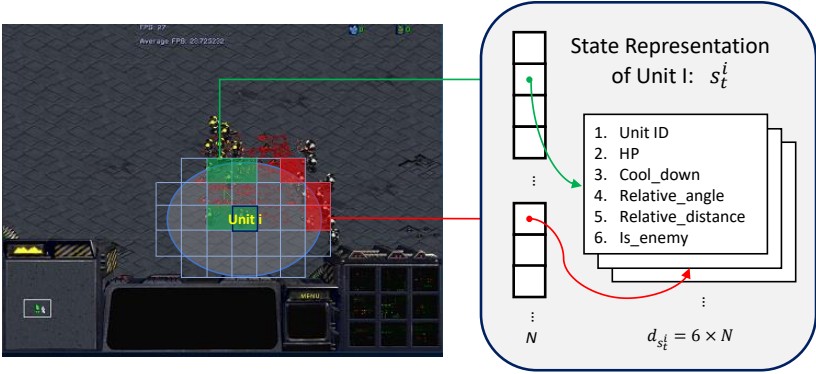

Figure 3: State definition of unit i in the task of $\{$15M vs. 16M, 10M vs. 13Z, 15W vs. 17W and 2D3Z$\}$

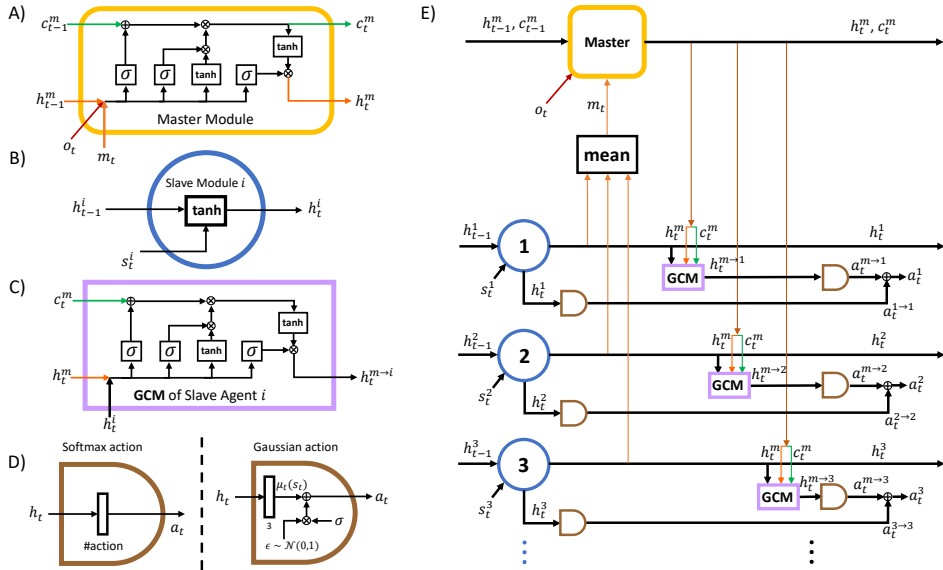

Figure 4: A) master module, B) slave module, C) gated composition module of slave i, D) Softmax/Gaussian action module and E) specific model architecture.

**State Features**    For the traffic junction task and the combat task, we just apply the original state features designed in [23]. For ''15M vs. 16M'', ''10M vs. 13Z'' and ''15W vs. 17W'', we adopt one representation similar to existing methods [19; 4; 5]. To be specific, the feature of an agent is a map of a certain-sized circle around it, as illustrated in Figure 3. If there is an agent in a cell, a feature vector shown in Fig. 3 will be included in the final observation map. A slightly different state definition is applied for the ''2D3Z'' task, where unit type is critical for learning correct collaborative policy. In practice, we directly add the type id as one component to the input state feature for the ''2D3Z'' task. Note that the proposed GCM module facilitates the learning of different strategy for different unit type, which is essential for winning this task.

**Action Definition**    For action definitions of the first two tasks, we use the default ones from [23]. Since ''15M vs. 16M'', ''10M vs. 13Z'' and ''15W vs. 17M'' needs a continuous action space. We apply the design from [19]. It contains a 3-dimension vector, each of which is of the range from -1 to 1, representing action type, relative angle and distance to the target respectively.

**Reward Definition**    The reward definition of the traffic junction task and the combat task is the same as in [23]. Here we define the reward for ''15M vs. 16M'', ''10M vs. 13Z'' and ''15W vs. 17W''

by simulating the design of that in the combat task. For each time step, there is a reward formulated in 2 to evaluate how the current action of agent i works.

$$r_t^i = \Delta n_{j \in N_m(i)}^t - \Delta n_{k \in N_e(i)}^t \tag{2}$$

Note that here $\Delta n_{j \in N_m(i)}^t = n_{j \in N_m(i)}^t - n_{j \in N_m(i)}^{t-1}$ (the definition of $\Delta n_{k \in N_e(i)}^t$ is similar) which is actually the change of the number of our units in the neighboring area. $N_m(i)$ and $N_e(i)$ refers to the sets containing our units and enemy units within a certain range of the current agent i. In experiments we define the range as a circle of a radius of 7 around agent i, which is exactly the circular range as 3 shows. Besides, we also define a terminal reward (as formulated in (3)) to add to the final reward at the end of each episode.

$$r_{terminal} = \begin{cases} 1 & if\ battle\ won \\ -0.2 & else \end{cases} \tag{3}$$

**Training Procedure** The entire training process is shown in Algorithm 1. Note that we follow the softmax policy as proposed in [23] for the first two discrete tasks. As for the cotinuous tasks {15M vs. 16M, 10M vs. 13Z, 15W vs. 17W}, a Gaussian policy is adopted for stochastic policy parameterization. Specifically, the output of the network is considered as the mean of a Gaussian policy distribution. By fixing the variance to $\sigma = 0.05$, the final actions are sampled as follows

$$a_t^i = \mu_t^i(s_t^i; \theta) + \sigma \cdot \mathcal{N}(0, 1), \tag{4}$$

And the score function in (1) can be computed as

$$\nabla_\theta \log \pi_\theta(s_t^i, a_t^i) = \frac{(a_t^i - \mu(s_t^i; \theta))}{\sigma^2} \frac{\partial \mu(s_t^i; \theta)}{\partial \theta}. \tag{5}$$

For the traffic junction task, we use a batch size of 16. For the combat task, a larger batch size of 144 is adopted for the training of both CommNet and our master-slave models. And for {15M vs. 16M, 10M vs. 13Z, 15W vs. 17W}, we find that a small batch size of 4 is good enough to guarantee a successful training. The learning rate is set to 0.001 for the first two tasks and 0.0005 for {15M vs. 16M, 10M vs. 13Z, 15W vs. 17W}. The action variance for {15M vs. 16M, 10M vs. 13Z, 15W vs. 17W} is initialized as 0.01 and drops down gradually as the training proceeds. For all tasks, the number of batch per training epoch is set to 100.

**Baselines** For comparison, we select three state-of-the-art MARL methods that have been tested on these tasks of interest. A brief introduction of these baselines are given as follow:

- **CommNet:** This method exploits a simple strategy for multi-agent communication. The idea is to gather a message output from all agents and compute the mean value. Then the mean value is spread to all agents again as an input signal of the next time step. In this way, a communication channel established for all agents. Such a simple strategy has been shown to work reasonably well for the first two tasks in [23] and for {15M vs. 16M, 10M vs. 13Z, 15W vs. 17W} when explored in [19].

- **GMEZO:** The full name of this method is GreedyMDP with Episodic Zero-Order Optimization. And it was proposed particularly to solve StarCraft micromanagement tasks in [27]. The main idea is to employ a greedy update over MDP for inter-agent communications, while combining direct exploration in the policy space and backpropagation.

- **BiCNet:** A bidirectional RNN is proposed to enable communication among multiple agents in [19]. The method adopts an Actor-Critic structure for reinforcement learning. It has been shown that their model is able to handle several different micromanagement tasks successfully as well as learn meaningful polices resembling skills of expert human players.

### 4.3 PERFORMANCE

Table 1 and Table 2 demonstrate the performance improvement of our method when compared with the baselines. For CommNet we directly run the released code on the traffic junction task and

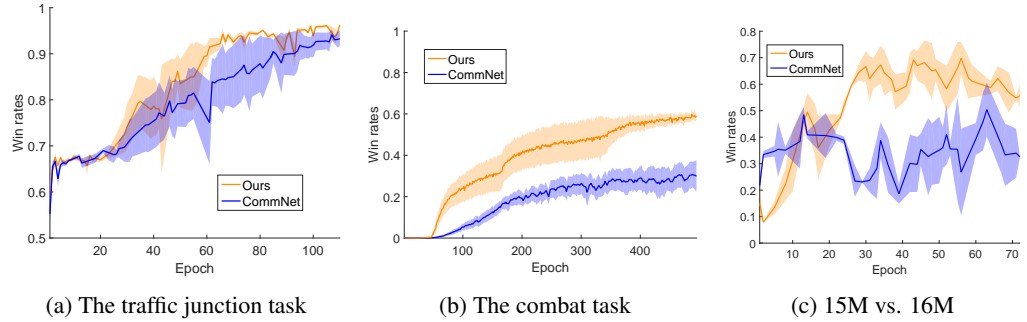

| (a) The traffic junction task | (b) The combat task | (c) 15M vs. 16M |

Figure 5: Comparing winning rates of different methods on all three tasks

the combat task using hyper-parameters provided in [23]. We compute the mean winning rates in Table 1 by testing the trained models for 100 rounds. However, since the code of GMEZO and BiCNet is not released yet, there is no report of their performance on traffic junction and combat tasks. Therefore we only compare with CommNet on these two tasks. And it can be seen that our MS-MARL model performs better than CommNet on both of the two tasks. To further clarify the contribution of the proposed master-slave scheme, we add another baseline "CommNet + Occupancy Map" which takes its original state as well as the occupancy map explicitly. As is shown, CommNet performs better with global information. However, with an explicit global planner (the "master" agent) designed in our model, such information seems better utilized to facilitate learning of more powerful collaborative polices.

The results on the more challenging StarCraft micromanagement tasks {15M vs. 16M, 10M vs. 13Z, 15W vs. 17W, 2D3Z} are displayed in Table 2. Obviously, on all the tasks, our MS-MARL method achieves consistently better performance compared to the available results of the three baselines.[2]

| Tasks | CommNet | CommNet + Occupancy Map | MS-MARL |
|---|---|---|---|
| Traffic Junction | $0.94 \pm 0.02$ | $0.95 \pm 0.01$ | $\mathbf{0.97 \pm 0.01}$ |
| Combat | $0.44 \pm 0.12$ | $0.49 \pm 0.01$ | $\mathbf{0.61 \pm 0.03}$ |

Table 1: Mean win rates of different methods on the first two discrete tasks

| Tasks | GMEZO | CommNet | BiCNet | MS-MARL |
|---|---|---|---|---|
| 15M vs. 16M | 0.79 | $0.69 \pm 0.04$ | 0.71 | $\mathbf{0.82 \pm 0.06}$ |
| 10M vs. 13Z | 0.57 | $0.44 \pm 0.02$ | 0.64 | $\mathbf{0.76 \pm 0.05}$ |
| 15W vs. 17W | 0.49 | $0.47 \pm 0.01$ | 0.53 | $\mathbf{0.60 \pm 0.05}$ |
| 2D3Z | 0.90 | $0.53 \pm 0.04$ | - | $\mathbf{0.92 \pm 0.01}$ |

Table 2: Mean win rates of different methods on the continuous StarCraft micromanagement tasks

Figure 5 shows the training process of CommNet and our MS-MARL method by plotting win rate curves for the first two tasks as well as "15M vs. 16M". As analyzed above, one key difference between our MS-MARL model and CommNet is that we explicitly facilitate an independent master agent reasoning with its own state and messages from all slave agents. From this plot, our MS-MARL model clearly enjoys better, faster and more stable convergence, which highlights the importance of such a design facilitating independent thinking.

---

[2]Note that the mean win rates of GMEZO, CommNet and BiCNet on the StarCraft tasks are all available from [19]. (Except for the win rate of CommNet on the task "2D3Z", which actually comes from our own implementation) However, it is notable that the performance of GMEZO reproduced by [19] is much lower compared to that in the original paper[27], which is probably caused by implementation difference. Therefore we directly follow the higher original results of GMEZO reported in [27] for comparison.

## 4.4 ABLATION ANALYSIS

**Analysis of Master/Slave Behaviours**

In the setting of the combat task, we further analyze how different components of our proposal contribute individually. Specifically, we compare the performance among the CommNet model, our MS-MARL model without explicit master state (e.g. the occupancy map of controlled agents in this case), and our full model with an explicit occupancy map as a state to the master agent. As shown in Figure 7 (a)(b), by only allowing an independently thinking master agent and communication among agents, our model already outperforms the plain CommNet model which only supports broadcasting the same message to all agents. Further more, by providing the master agent with its unique state, our full model finally achieves a significant improvement over the CommNet model. Note here that, every information revealed from the extra occupancy map is by definition already included to each agents state as their locations.

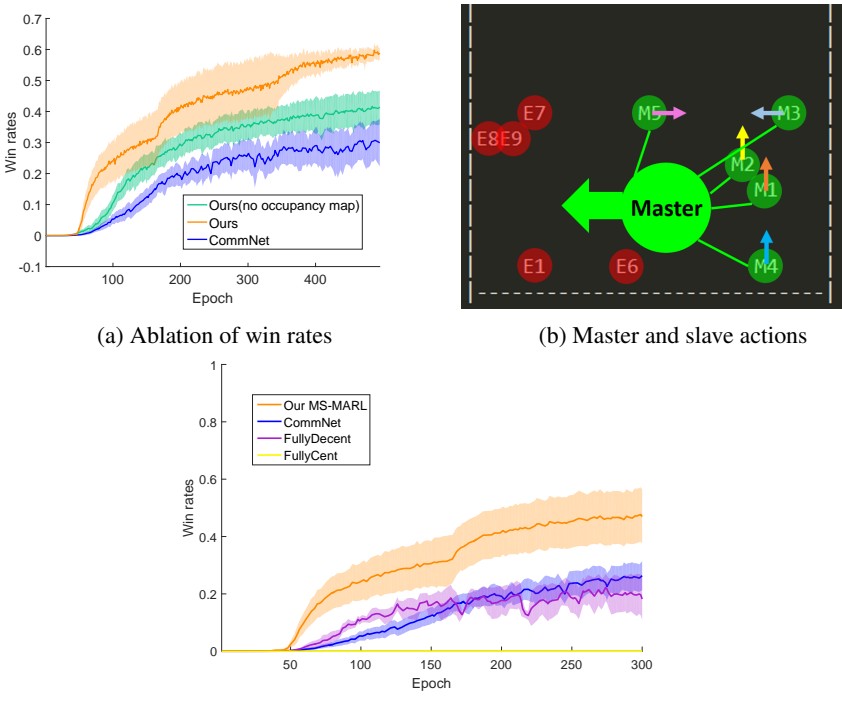

(a) Ablation of win rates        (b) Master and slave actions

(c) Ablation of different centralization level

Figure 6: Ablation study

Another interesting observation is how the master agent and each of the slave agents contribute to the final action choices (as shown in Fig. 6). We observe that the master agent does often learn an effective global policy. For example, the action components extracted from the master agent lead the whole team of agents move towards the enemies regions. Meanwhile, all the slave agents adjust their positions locally to gather together. This interesting phenomenon showcases different functionality of different agent roles. Together, they will gather all agents and move to attack the enemies, which seems to be quite a reasonable strategy in this setting.

**Comparison of Centralization Level**

Our method combines the advantages of both centralized and decentralized perspectives. To better demonstrate this point, we compare a set of baselines by modifying the structure of our model. These baselines include: 1. FullyCent, 2. FullyDecent, 3. Our MS-MARL, 4. CommNet. Specifically, FullyCent has only a single master agent controlling all slaves. And the FullyDecent refers to a scenario that all slave agents play independently with each other without any communication. Note that FullyDecent can also be considered as taking other agents as part of the environment, and thus the slave agents communicate implicitly by interacting with the environment. This setting is

widely used in [5][14][26]. We also include CommNet here since it can be seen as a midstage from FullyDecent to MS-MARL. In Figure 6 (c) we draw the training curve of the first 300 steps for all the baselines. It is evident that our MS-MARL performs better than the other baselines with a quite clear margin(more than 20 percent higher than the second best one). Also we find that it is very hard for a single master agent to control all units (unfortunately it earns a 0% winning chance in this case). While independent agents do learn some winning policy, their performance is far lower from MS-MARL but comparable to CommNet. From these results, we can see that a stronger global planner as well as a more powerful gating mechanism for message collection/dispatching is necessary for multi-agent coordination.

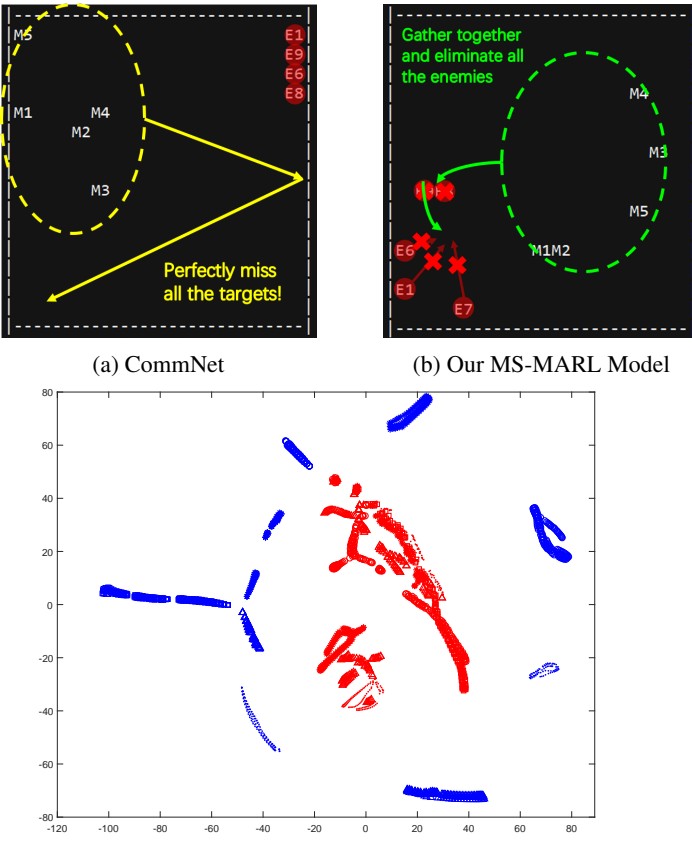

(a) CommNet        (b) Our MS-MARL Model

(c) Visualization of master agent's hidden state (blue: full version of our MS-MARL model, red: original version without occupancy map or GCM module)

Figure 7: Analysis of learned policies: (a) a failure case that CommNet misses the targets (b) a successful case of our MS-MARL model (c) Visualizing the master's hidden states of two versions of our MS-MARL model

## 4.5 ANALYSIS OF LEARNED POLICIES

Note that the CommNet model has already been verified to have learned meaningful policies in these tasks according to [23; 19]. However, in our experiments, we often observe more successful policies learned by our method which may not be captured very well by the CommNet model.

For example, in the case of combat task, we often observe that some agents of the CommNet model just fail to find the enemies (potentially due to their limited local views and ineffective communications) and therefore lose the battle in a shorthanded manner, see e.g. Figure 7 (a). As a contrast, the learned model of our MS-MARL method usually gathers all agents together first and then moves them to attack the enemies. Figure 7 (b) showcases one example.

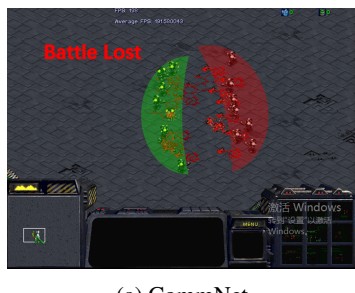 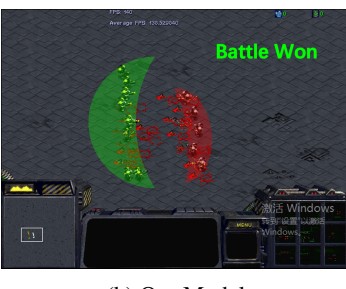

(a) CommNet                              (b) Our Model

Figure 8: Illustration of the policy learned by our MS-MARL method: (a) is a failure case of Comm-Net (b) showcases the successful "Pincer Movement" policy learned by our MS-MARL model

Another support case is in the task of "15M vs. 16M". In this task, our model learns a particular policy of spreading the agents into a half-moon shape ("Spread Out") to focus fire and attacking the frontier enemies before the others enter the firing range, as demonstrated in Fig. 8 (b). Actually, this group behavior is similar to the famous military maneuver "Pincer Movement" which is widely exploited in representative battles in the history. Although CommNet sometimes also follows such kind of policy, it often fails to spread to a larger "pincer" to cover the enemies and therefore loses the battle. Figure 8 (a) shows one of such examples. The size of the "pincer" seems especially important for winning the t ask of "15M vs. 16M" where we have less units than the enemies.

From the above analysis, we see that the master module does contribute to a higher performance. To further explore the key ingredients, we generate a visualization of the hidden state from the master agent (projected to 2D with t-SNE[11], shown in Figure 7 (c)). Specifically, we plot the hidden state of two different versions of our MS-MARL model: the blue points version is coupled with a GCM module as well as occupancy map as master's input (with a 61% win rate), while the red points are just from the original version (with only a 55% win rate). Although an explicit explanation of the meanings of the hidden states cannot be certified at this point, what excites us here is that the distribution of master's message does correspond to several representative successful polices. This interesting phenomenon indicates that the hidden states of the master agent effectively carry meaningful messages, according to which, the distribution of successful policies stands out from that of relatively poor policies.

## 5 CONCLUSION

In this paper, we revisit the master-slave architecture for deep MARL where we make an initial stab to explicitly combine a centralized master agent with distributed slave agents to leverage their individual contributions. With the proposed designs, the master agent effectively learns to give high-level instructions while the local agents try to achieve fine-grained optimality. We empirically demonstrate the superiority of our proposal against existing MARL methods in several challenging mutli-agent tasks. Moreover, the idea of master-slave architecture should not be limited to any specific RL algorithms, although we instantiate this idea with a policy gradient method, more existing RL algorithms can also benefit from applying similar schemes.

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
