# OpenReview forum: "Revisiting The Master-Slave Architecture In Multi-Agent Deep Reinforcement Learning"
_ICLR.cc/2018/Conference — Reject_

### Official Review · AnonReviewer2 · 2017-11-23
**results seem nice, but novelty less clear**

**Rating:** 5
**Confidence:** 5

**Review:**

This paper investigates multiagent reinforcement learning  making used of a "master slave" architecture (MSA). On the positive side, the paper is mostly well-written, seems technically correct, and there are some results that indicate that the MSA is working quite well on relatively complex tasks. On the negative side, there seems to be relatively limited novelty: we can think of MSA as one particular communication (i.e, star) configuration one could use is a multiagent system. One aspect does does strike me as novel is the "gated composition module", which allows differentiation of messages to other agents based on the receivers internal state. (So, the *interpretation* of the message is learned). I like this idea, however, the results are mixed, and the explanation given is plausible, but far from a clearly demonstrated answer.

There are some important issues that need clarification:

* "Sukhbaatar et al. (2016) proposed the “CommNet”, where broadcasting communication channel among all agents were set up to share a global information which is the summation of all individual agents. [...] however the summed global signal is hand crafted information and does not facilitate an independently reasoning master agent."
-Please explain what is meant here by 'hand crafted information', my understanding is that the f^i in figure 1 of that paper are learned modules?
-Please explain what would be the differences with CommNet with 1 extra agent that takes in the same information as your 'master'.


*This relates also to this:

"Later we empirically verify that, even when the overall in-
formation revealed does not increase per se, an independent master agent tend to absorb the same
information within a big picture and effectively helps to make decision in a global manner. Therefore
compared with pure in-between-agent communications, MS-MARL is more efficient in reasoning
and planning once trained. [...]
Specifically, we compare the performance among the CommNet model, our
MS-MARL model without explicit master state (e.g. the occupancy map of controlled agents in this
case), and our full model with an explicit occupancy map as a state to the master agent. As shown in
Figure 7 (a)(b), by only allowed an independently thinking master agent and communication among
agents, our model already outperforms the plain CommNet model which only supports broadcast-
ing communication of the sum of the signals."

-Minor: I think that the statement "which only supports broadcast-ing communication of the sum of the signals" is not quite fair: surely they have used a 1-channel communication structure, but it would be easy to generalize that.

-Major: When I look at figure 4D, I see that the proposed approach *also* only provides the master with the sum (or really mean) with of the individual messages...? So it is not quite clear to me what explains the difference.


*In 4.4, it is not quite clear exactly how the figure of master and slave actions is created. This seems to suggest that the only thing that the master can communicate is action information? It this the case?

* In table 2, it is not clear how significant these differences are. What are the standard errors?

* The section 3.2 explains standard things (policy gradient), but the details are a bit unclear. In particular, I do not see how the Gaussian/softmax layers are integrated; they do not seem to appear in figure 4?

* I cannot understand figure 7 without more explanation. (The background is all black - did something go wrong with the pdf?)




Details:
* references are wrongly formatted throughout.

* "In this regard, we are among the first to combine both the centralized perspective and the decentralized perspective"
This is a weak statement (E.g., I suppose that in the greater scheme of things all of us will be amongst the first people that have walked this earth...)


* "Therefore they tend to work more like a commentator analyzing and criticizing the play, rather than
a coach coaching the game."
-This sounds somewhat vague. Can it be made crisper?

* "Note here that, although we explicitly input an occupancy map to the master agent, the actual infor-
mation of the whole system remains the same."
This is a somewhat peculiar statement. Clearly, the distribution of information over the agents is crucial. For more insights on this one could refer to the literature on decentralized POMDPs.

---

> ### Author Response · Authors · 2018-01-05
> **Response to review**
>
> There are two key differences from CommNet: 1) CommNet pass back the sum of all agents’ hidden state directly to each agent at the next time step (this direct pass is what we meant by handcrafted information); whereas our applied an LSTM to process such information from time to time and therefore formulates an independently thinking master agent; 2) when passing back the processed information, we applied a GCM unit to effectively merge information/“thoughts” from both the master agent and the slave agents. Both independent thinking LSTM and the GCM merging module are learnable and their effectiveness has been shown when compared to the CommNet model. We highly recognize CommNet and our proposal was directly motivated from CommNet. We have refined some of the statement to avoid misleading.
>
> * This seems to suggest that the only thing that the master can communicate is action information? It this the case?
>
> This is not the case. The master will pass along its hidden states to the GCM module which will process such information and that from the slave agents to finally output actions.
>
> * What are the standard errors?
>
> Standard errors are provided in the updated tables.
>
> * The section 3.2 explains standard things (policy gradient), but the details are a bit unclear. In particular, I do not see how the Gaussian/softmax layers are integrated; they do not seem to appear in figure 4?
>
> The Gaussian/softmax layers are actually after the hidden state output of the GCM module (to generate master's action) and each slave agent's RNN module (to generate slave's action).
>
> * I cannot understand figure 7 without more explanation. (The background is all black - did something go wrong with the pdf?)
>
> Thanks for your reminding. The black background is due to the command line output of the combat task environment. As a simple game environment based on MazeBase (Sukhbaatar et al. 2015), the visualization of this task is printed to command line as it was originally implemented. Thus the two examples are obtained by directly cropping command line output. The yellow and green dashes/arrows are added by us to illusrate the behaviours of agents. We will try to replace this demonstration to a more reader-friendly version in the camera-ready version if accepted.
>
> And the details suggestions are very helpful,  we fixed them in the updated version. Thanks.

---

> > ### Comment · AnonReviewer2 · 2018-01-11
> > **...**
> >
> > I am not sure if this really clarifies much.
> >
> > "CommNet pass back the sum of all agents’ hidden state directly to each agent at the next time step (this direct pass is what we meant by handcrafted information); whereas our applied an LSTM to process such information from time to time and therefore formulates an independently thinking master agent"
> >
> > -I am not sure if I understand correctly, but I think this says "we use LSTMs rather than multilayer feedforward networks for the components f_i", is that correct? (Fig. 1 clearly shows that you do communicate at every timestep, I am not sure why this is less handcrafted)
> >
> > "we applied a GCM"
> > -Yes, I did recognize the GCM as a novel feature in my review. Perhaps its merit could be investigated by comparing "with CommNet with 1 extra agent that takes in the same information as your 'master'." ?
> >
> > I don't understand why the Gaussian/softmax layers are not clarified in the updated paper. "on the top layer" is just not so clear.
> >
> > The standard errors look good, as such the evaluation does imply that there are merits to the overall approach. However, I still think that in the current form, it is not quite clear exactly what explains this difference in performance.

---

> > > ### Author Response · Authors · 2018-01-17
> > > **Response**
> > >
> > > Thanks for the updates.
> > >
> > > We have updated the argument about differences from CommNet as follows, emphasizing the independently and recurrently reasoning property of the master agent.
> > >
> > > "This design represents an initial version of the proposed master-slave framework, however it does not facilitate an independently reasoning master agent which takes in messages from all agents step by step and processes such information in an recurrent manner."
> > >
> > >
> > > Taking the advice regarding the master's global information, we add another baseline "CommNet + OccupancyMap" which takes its original state as well as the occupancy map explicitly. The results show that CommNet performs better with this additional global information. However, with an explicit global planner (the "master" agent) designed in our model, such information seems better utilized to facilitate learning of more powerful collaborative polices. More details please refer to Table 1 and the comments in the updated paper. Regardless, note that, all information revealed from the extra occupancy map is by definition already included in all agents' state. To be specific, the occupancy map only includes the positions and IDs of our units without any of the enemies. And it is mentioned in the original CommNet paper* section 4.3.1 and 4.3.3 that the absolute location of each agent is already included in its input state features. Thus the occupancy map adds no extra information to the total information of all agents, but organizes them in an explicit global manner.
> > >
> > > Softmax/Gaussian action layers are added to Figure 4.
> > >
> > > We sincerely appreciate the constructive suggestions.
> > >
> > >
> > > *Sainbayar Sukhbaatar, Rob Fergus, et al. Learning multiagent communication with backpropagation. In Advances in Neural Information Processing Systems, pp. 2244–2252, 2016.

---

### Official Review · AnonReviewer1 · 2017-11-25

**Rating:** 5
**Confidence:** 4

**Review:**

The paper proposes a neural network architecture for centralized and decentralized settings in multi-agent reinforcement learning (MARL) which is trainable with policy gradients.
Authors experiment with the proposed architecture on a set of synthetic toy tasks and a few Starcraft combat levels, where they find their approach to perform better than baselines.

Overall, I had a very confusing feeling when reading the paper. First, authors do not formulate what exactly is the problem statement for MARL. Is it an MDP or poMDP? How do different agents perceive their time, is it synchronized or not? Do they (partially) share the incentive or may have completely arbitrary rewards?
What is exactly the communication protocol?

I find this question especially important for MARL, because the assumption on synchronous and noise-free communication, including gradients is too strong to be useful in many practical tasks.

Second, even though the proposed architecture proved to perform empirically better that the considered baselines, the extent to which it advances RL research is unclear to me.
Currently, it looks

Based on that, I can’t recommend acceptance of the paper.

To make the paper stronger and justify importance of the proposed architecture, I suggest authors to consider relaxing assumptions on the communication protocol to allow delayed and/or noisy communication (including gradients).
It would be also interesting to see if the network somehow learns an implicit global state representation used for planning and how is the developed plan changed when new information from one of the slave agents arrives.

---

> ### Author Response · Authors · 2018-01-05
> **Response to review**
>
> The overall problem statement of the MARL problem we consider here should be formulated as an MDP (supposed we do not take “fog of war” etc. into account).
>
> Although each agent can only observe the environment partially, when considering the global optimal planning, we should view these problems from a centralized perspective and formulate it as one global MDP where the action of such MDP is defined as the concatenation of all agents’ actions. However since such an action space can be very large when the number of agents increases, therefore people tend to apply decentralized perspective, where each agent can be viewed as having its own MDP and than try to implement certain communication protocols between agents to facilitate collaborations in-between. In this regard, our work proposed to apply a master-slave communication architecture and to realize such architecture with learnable neural networks so that the final communication protocols are learned from many times of RL try and errors in the same way as the parameters of any RL models.
>
> Technically speaking, we have no assumption of synchronous and noise-free communication. Regarding communication, as mentioned above, since the final protocols will be driven by training data, we almost do not have any assumptions except for pre-defined the certain hyper-parameters such as the dimensionality etc. As for synchronousness, our model does not make such an assumption, e.g. the master agent can take empty or zero values if any of the slave agents have no output at certain time steps.
>
> Regarding advances to RL research, we provide a novel practical MARL solution which encodes the idea  of master-slave communication architecture. There are currently few theoretical results in this paper, especially on properties of the global MDP statements. However, motivated from the concept of combining both the centralized and decentralized perspectives of MARL, we are working on another draft explaining how effective communications between agents helps finding better optima of the global MDP.
>
> As for implicit global state representations, this is a very good suggestion, we have added some analysis where we tried to visualize and understand the hidden state of the master agent. More details can be found in sec 4.5.

---

### Official Review · AnonReviewer3 · 2017-11-28
**Somewhat encouraging results on StarCraft, method is incremental, paper needs work**

**Rating:** 4
**Confidence:** 3

**Review:**

The paper presents results across a range of cooperative multi-agent tasks, including a simple traffic simulation and StarCraft micro-management. The architecture used is a fully centralized actor (Master) which observes the central state in combination with agents that receive local observation, MS-MARL.
A gating mechanism is used in order to produce the contribution from the hidden state of the master to the logits of each agent.  This contribution is added to the logits coming from each agent.

Pros:
-The results on StarCraft are encouraging and present state of the art performance if reproducible.

Cons:
-The experimental evaluation is not very thorough:
No uncertainty of the mean is stated for any of the results. 100 evaluation runs is very low. It is furthermore not clear whether training was carried out on multiple seeds or whether these are individual runs.

-BiCNet and CommNet are both aiming to learn communication protocols which allow decentralized execution. Thus they represent weak baselines for a fully centralized method such as MS-MARL.
The only fully centralized baseline in the paper is GMEZO, however results stated are much lower than what is reported in the original paper (eg. 63% vs 79% for M15v16). The paper is missing further centralized baselines.

-It is unclear to what extends the novelty of the paper (specific architecture choices) are required. For example, the gating mechanism for producing the action logits is rather complex and seems to only help in a subset of settings (if at all).

Detailed comments:
"For all tasks, the number of batch per training epoch is set to 100."
What does this mean?

Figure 1:
This figure is very helpful, however the colour for M->S is wrong in the legend.

Table 2:
GMEZO win rates are low compared to the original publication.
What many independent seeds where used for training? What are the confidence intervals? How many runs for evaluation?


Figure 4:
B) What does it mean to feed two vectors into a Tanh? This figure currently very unclear. What was the rational for choosing a vanilla RNN for the slave modules?

Figure 5:
a) What was the rational for stopping training of CommNet after 100 epochs? The plot looks like CommNet is still improving.
c) This plot is disconcerting. Training in this plot is very unstable. The final performance of the method ('ours') does not match what is stated in 'Table 2'. I wonder if this is due to the very small batch size used ("a small batch size of 4 ").

---

> ### Author Response · Authors · 2018-01-05
> **Response to review**
>
> We appreciate the positive comments and address the concerns as follows.
>
> Regarding experimental evaluation, thanks to the suggestion, we have added uncertainty results aka std in the latest tables. The training was carried out on multiple seeds and we chose the best 5 runs and computed the statistics (which is somehow standard since training of RL is not always stable).
>
> The suggestion to compare with more centralized baselines is enlightening, we manage to realized a simple centralized baseline based on one modification from our own model. More detailed results are provided in the newly section 4.4. In a nutshell, this method performs rather poorly probably due to scalability issues, but we agree there are still room to explore in this regard. Regarding the performance of GMEZO, we have originally reported the reproduced results from the BicNet paper for consistency considerations, in the updated version we clarify this point and have now reported the highest ones from both. Still ours are consistently superior.
>
> It is slightly subjective when discussing how novel a proposal is. What we can confirm is that we are the first to explicitly explore the master-slave communication architecture for MARL and effectively show its superiority to existing communication proposals such as that in BicNet and CommNet. The proposed GCM is a novel communication processor. Although the initial experimental results didn’t strongly justify its advantages, we have tried our more challenging settings such as heterogeneous starcraft tasks etc. We included one such case and related analysis in the updated version. We agree and has never denied that master-salve and LSTM cells are existing wisdoms, our major contribution is the novel and effective instantiation in the MARL settings, out of which we reported very promising performance on challenging tasks such as starcraft micromanagements.
>
> All detailed comments are very helpful, we have properly addressed them in the updated version.

---

### Author Response · Authors · 2018-01-05
**Updates to the paper**

We summarize the updates made to the paper as follow:

1. Add one more StarCraft micromanagement task (both task description and new evaluation results etc.) "Dragoons vs. Zealots" where heterogeneous agents are involved.

2. A new experiment is included in section 4.4 to compare models of different centralization level (both the curves in Figure 6 (c) and the related analysis).

3. Add a visualization of the master agent's hidden state to demonstrate the effectiveness of occupancy map and GCM module in our proposal (and the related analysis).

4. Update table 1 and table 2 with standard errors, follow the results of GMEZO reported in the original paper.

5. Fix some typos and details according to reviewers' suggestions.

6. Statement about CommNet has been revised in Section 2.

7.  Results of a new baseline "CommNet + Occupancy Map" are provided in Table 1.

8. Softmax/Gaussian action layers have been added to Figure 4.

---

### Decision · Program_Chairs · 2018-01-29
**ICLR 2018 Conference Acceptance Decision**

**Decision:**

Reject

**Comment:**

The authors present a centralized neural controller for multi-agent reinforcement learning.   The reviewers are are not convinced that there is sufficient novelty, considering the authors setup as essentially a special case of other recent works, with added adjustments to the neural-networks that are standard in the literature.

I personally am more bullish about this paper than the reviewers, as I think engineering an architecture to perform well in interesting scenarios is worth reporting.  However, the reviewers are mostly in agreement, and their reviews were neither sloppy nor factually incorrect.  So I will recommend rejection, following their judgement.

Nevertheless, I encourage the authors to continue strengthening the results and the presentation and resubmit.